# Unified divergent strategy towards the total synthesis of the three sub-classes of hasubanan alkaloids

Guang Li[1], Qian Wang[1] & Jieping Zhu [1✉]

Elegant asymmetric synthesis of hasubanan alkaloids have been developed over the past decades. However, a divergent approach leading to all three sub-classes of this family of natural products remains unknown. We report herein the realization of such an endeavor by accomplishing enantioselective total syntheses of four representative members. The synthesis is characterized by catalytic enantioselective construction of the tricyclic compounds from which three different intramolecular C-N bond forming processes leading to three topologically different hasubanan alkaloids are developed. An aza-Michael addition is used for the construction of the aza-[4.4.3]-propellane structure of (-)-cepharamine, whereas an oxidation/double deprotection/intramolecular hemiaminal forming sequence is developed to forge the bridged 6/6/6/6 tetracycle of (-)-cepharatines A and C and a domino bromination/double deprotection/cyclization sequence allows the build-up of the 6/6/5/5 fused tetracyclic structure of (−)-sinoracutine.

[1] Laboratory of Synthesis and Natural Products, Institute of Chemical Sciences and Engineering, Ecole Polytechnique Fédérale de Lausanne, EPFL-SB-ISIC-LSPN, BCH5304, 1015 Lausanne, Switzerland. ✉email: jieping.zhu@epfl.ch

The bridged tetracycle **A**, particularly its enantiomer *ent*-**A**, is a core structure of morphinan alkaloids such as (-)-salutaridine (**1**) and (-)-morphine (**2**) (Fig. 1a)[1]. Interestingly, a 1,2-shift of the C-N bond from C9 to C14 produces an aza-[4.4.3]-propellane[2] structure **B** that constitutes a basic skeleton of the hasubanan alkaloids[3], with (-)-hasubanonine (**3**)[4] and (-)-cepharamine (**4**)[5] being representative examples. Recently, the structural diversity of hasubanan alkaloids has been further expanded by the isolation of natural products bearing the tetracyclic skeletons of types **C** and **D**. Thus cepharatines A (**5**), C (**6**), B (**7**), and D (**8**) bearing a bridged piperidine ring were isolated from *Stephania cepharantha*[6], while sinoracutine (**9**)[7] and sinoraculine (**10**)[8] with a 6/6/5/5 tetracyclic structure were isolated from *Sinomenium acutum* and *S. cepharantha* of the Menispermaceae family, respectively. (-)-Sinoracutine (**9**) displays potent antioxidant activity that could potentially be exploited as a lead in searching for drugs against neurodegenerative diseases. Biosynthetically, morphinan and hasubanan alkaloids are derived from the intramolecular oxidative coupling of the corresponding tetrahydroisoquinoline. For example, both morphine (**2**) and hasubanonine (**3**) are biosynthesized from (-)- or (+)-salutaridine (**1**), which is in turn obtained by intramolecular oxidative coupling of (*R*)- or (*S*)-reticuline (**11**) via formation of C12-C13 bond of the radical intermediate **12** (Fig. 1b)[1,3,9].

For their structural resemblance to morphinan alkaloids, the hasubanan alkaloids have attracted attention of synthetic chemists for over half a century. Since the first elegant total syntheses of (±)-hasubanonine (**3**), (±)-cepharamine (**4**), and (±)-metaphanine by Ibuka in early 1970s[10–12], innovative strategies have been developed to construct the unique propellane structure of this important family of natural products[13–25]. However, the first asymmetric total synthesis was accomplished only in 1998 by Schultz and Wang using a chiral auxiliary approach[26]. A dozen years later, the Herzon group developed in 2011 an efficient asymmetric synthesis of both the hasubanan[27,28] and the acutumine families of alkaloids[29–31]. Diastereoselective addition of alkynyl lithium **13** to *N*-methyl iminium **14** at −90 °C afforded key intermediate **15** as a single detectable diastereomer (Fig. 1c). Concurrently, Reisman and co-workers designed a general strategy involving a diastereoselective addition of Grignard reagent **16** to chiral *N*-*tert*-butanesulfinimine **17** for the generation of C14 tetrasubstituted stereocenter in **18** (Fig. 1d)[32]. A divergent synthesis of (-)-8-demethoxyrunanine and (-)-cepharatines A (**5**), C (**6**), and D (**8**), belonging to the hasubanan subgroups **B** and **C**, respectively, were successfully realized from the common intermediate **18**. More recently, Kim's group completed the total synthesis of (-)-runanine and the formal synthesis of (-)-8-demethoxyrunanine as well as (−)-cepharatine D (**8**)[33]. Treatment of enantioenriched chiral amino ester **19** (*ee* 97%) with KO*t*Bu at 0 °C afforded enolate **20**, which underwent 5-*exo* cyclization to afford pyrrolidine **21** (*ee* 95%) with a memory of chirality effect (Fig. 1e). The common feature of these three approaches is the installation of the chiral α-tertiary amine (C14) at the beginning of the synthesis. A completely different strategy implicating the generation of C13 quaternary carbon stereocenter was designed by Trauner and co-workers for the total synthesis of (-)-sinoracutine (**9**)[34]. Thus oxa-Michael addition of **22** to phenyl vinyl sulfoxide (**23**) afforded **24** (NaH, THF, then KH, 0 °C to rt), which, upon heating in 1,2-dichlorobenzene (NaHCO₃, 176 °C), underwent *syn*-elimination of sulfenic acid to afford the chiral allylic enol ether. The Claisen rearrangement of the latter furnished then the aldehyde **25** with perfect chirality transfer from C6 to C13 (Fig. 1f). Emanating from this study, a plausible mechanism responsible for the facile racemization of (-)-sinoracutine was proposed to rationalize the discrepancy in the reported optical rotation value of this structurally unique natural product. To the best of our knowledge, no divergent

total synthesis of all three sub-types of hasubanan alkaloids (**B**, **C**, **D**, Fig. 1a) has been reported to date.

Here we report a unified strategy toward the total synthesis of the three sub-classes of hasubanan alkaloids in continuation with our research program dealing with the development of divergent synthesis of skeletally diverse natural products[35–38]. The retrosynthetic analysis featuring an early stage creation of the C13 quaternary stereocenter is shown in Fig. 2. In a forward sense, sinoracutine (**9**) would be accessed by way of an intramolecular α-amination of the fused cyclopentenone **26** (*n* = 1) via the formation of C5-N bond (pathway a). From the cyclohexenone **27** (*n* = 2), an intramolecular aza-Michael addition would forge the C14-N bond leading to the aza-[4.4.3]-propellane skeleton of cepharamine (**4**, pathway b), whereas a sequence of oxidation to α-diketone followed by deprotection and intramolecular hemiaminal formation would generate cepharatines A (**5**) and C (**6**) via the formation of C6-N (pathway c). Both **26** and **27** could be synthesized by functional group manipulation of **28**, available by enantioselective dearomatizative Michael addition of α-substituted β-naphthol **29** to nitroethylene (**30**) developed by You and co-workers[39]. In addition to the divergency of the synthesis, the catalytic enantioselective generation of the quaternary C13 stereocenter differentiates also the present strategy from other reported strategies.

## Results

**Catalytic asymmetric synthesis of dihydronaphthalen-2-one.** The synthesis began with the known aldehyde **31**, easily prepared from isovanillin (Fig. 3)[40]. Wacker oxidation of **31** under standard conditions followed by a base-promoted aldol condensation and subsequent aromatization furnished the β-naphthol **32** in 73% yield[41]. Allylation of **32** with allyl bromide afforded **33**, which, upon refluxing in 1,2-dichlorobenzene, underwent a regioselective Claisen rearrangement to provide α-allyl-β-naphthol **29** in 86% overall yield[42]. Following You's procedure[39], dearomatizative Michael addition of β-naphthol **29** to nitroethylene (**30**) in the presence of Takemoto's (1 *R*,2 *R*)-thiourea catalyst **34**[43] afforded α,α-disubstituted β-naphthalenone **28** in 66% yield with 93% *ee* (See Supplementary Information). The absolute configuration of C13 of compound **28** was tentatively assigned based on You's report and was later confirmed by the synthesis of the target molecules. A 1,4-reduction of enone with L-selectride at −78 °C delivered **35** in 92% yield[44]. Further reduction of **35** with LiAlH₄ afforded the amino alcohol, which, without purification, was chemoselectively *N*-acylated with methyl chloroformate under Schotten–Baumann conditions to provide the methyl carbamate **36**. Reduction of **36** with LiAlH₄ afforded the *N*-methylamine **37**. *N*-Boc protection of **37** followed by in situ oxidation of the secondary alcohol to ketone furnished **38**, a common intermediate in our planned synthesis of (-)-sinoracutine (**9**), (-)-cepharamine (**4**), (-)-cepharatines A (**5**), and C (**6**). Compound **38** was obtained in 68% overall yield from **35** (4 steps) without purification of any intermediate. Due to the proximity between the nitro and carbonyl groups, the redox manipulation of the ketone function is needed to allow the smooth and high yield conversion of the nitro to the protected amino group.

**Total synthesis of (-)-sinoracutine (9).** With the intermediate **38** in hands, we directed our attention toward the synthesis of sinoracutine (**9**) (Fig. 4). The Wacker oxidation of **38** provided diketone intermediate **39** in 78% yield. Performing the intramolecular aldol condensation of **39** in *t*BuOH in the presence of potassium *tert*-butoxide (*t*BuOK) proceeded slowly affording tricyclic enone **26** in low yield together with multiple unidentified

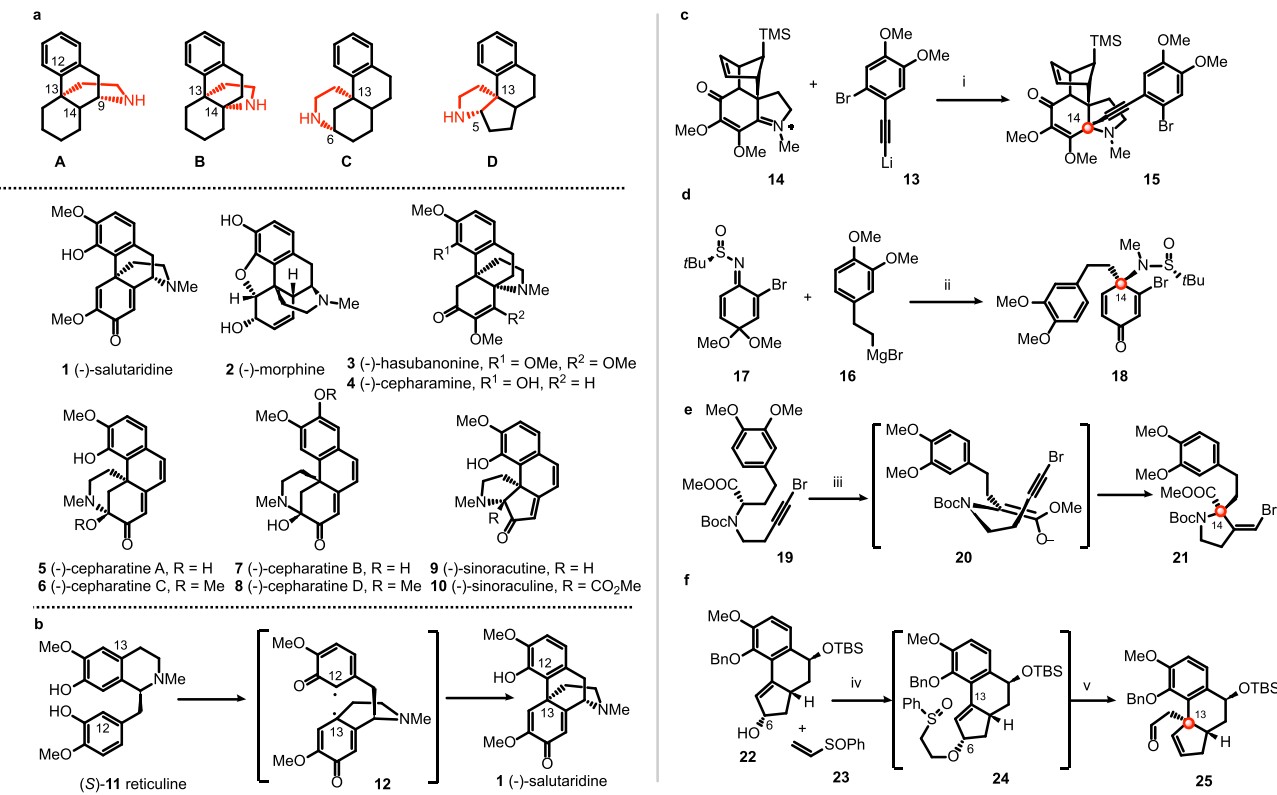

**Fig. 1 Hasubanan alkaloids and reported asymmetric syntheses. a** Representative skeletons of morphinan (**A**) and hasubanan alkaloids (**B**, **C**, and **D**); **b** biosynthesis of morphinan and hasubanan alkaloids; **c** Herzon's synthesis: construction of α-tertiary amine by diastereoselective nucleophilic addition of arylacetylide to chiral iminium salt; **d** Reisman's synthesis: construction of α-tertiary amine by diastereoselective addition of Grignard reagent to chiral *N*-*tert*-butanesulfinimine; **e** Kim's synthesis: construction of α-tertiary amine from α-amino ester via the memory of chirality approach; **f** Trauner's synthesis: construction of quaternary carbon stereocenter by stereoselective [3,3]-sigmatropic rearrangement. Reagents and conditions: (i) −90 °C, 62%; (ii) THF, −78 °C, then MeI, HMPA, then aq. AcOH, 77%, d.r. 96:4; (iii) KO*t*Bu, DMF, 0 °C, 79%; (iv) NaH, THF, then KH, 0 °C to rt; (v) NaHCO₃, 1,2-dichlorobenzene, 176 °C, 57% over 2 steps. HMPA hexamethylphosphoramide, DMF *N,N*-dimethylformamide, THF tetrahydrofuran.

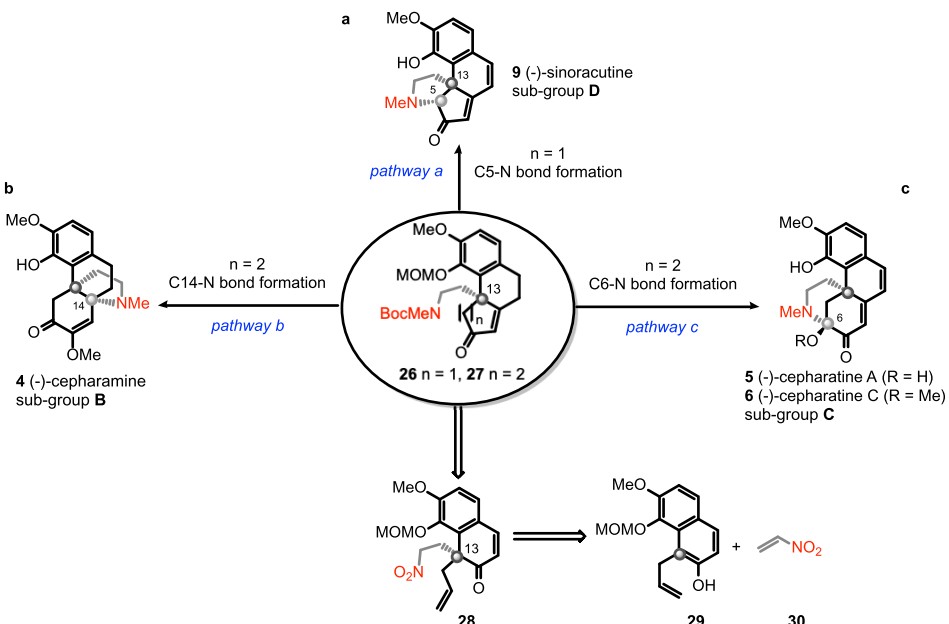

**Fig. 2 Retro-synthetic analysis of hasubanan alkaloids: a unified synthetic route. a** Pathway a: formation of C5-N bond for the synthesis of sinoracutine (**9**) of sub-group **D**; **b** Pathway b: formation of C14-N bond for the synthesis of (-)-cepharamine (**4**) of sub-group **B**; **c** Pathway c: formation of C6-N bond for the synthesis of (-)-cepharatines A (**5**) and C (**6**) of sub-group **C**.

**Fig. 3 Catalytic enantioselective synthesis of 1,1-disubstituted dihydronaphthalen-2-one (38).** Reagents and conditions: (a) PdCl$_2$ (0.05 equiv), CuCl$_2$ (1.5 equiv), O$_2$, DMF/H$_2$O = 7/1, rt, then NaOH (4.0 equiv), MeOH, 73%; (b) Allyl bromide (1.5 equiv), K$_2$CO$_3$ (1.5 equiv), acetone, reflux, 98%; (c) 1,2-Dichlorobenzene, 180 °C, 86%; (d) (1R,2R)-Takemoto thiourea catalyst **34** (0.05 equiv), nitroethylene (**30**, 2.0 equiv), 3 Å MS, rt, 66%, 93% ee; (e) L-Selectride (1.2 equiv), THF, −78 °C, 92%; (f) LiAlH$_4$ (5.0 equiv), Et$_2$O, 0 °C to rt; (g) ClCO$_2$Me (2.0 equiv), Na$_2$CO$_3$ (3.0 equiv), THF/H$_2$O = 3/1, rt; (h) LiAlH$_4$ (5.0 equiv), THF, reflux; (i) Boc$_2$O (1.0 equiv), DCM, rt, then DMP (1.0 equiv), NaHCO$_3$ (5.0 equiv), 0 °C to rt, 68% yield from **35**. DCM dichloromethane, DMP Dess–Martin periodinane.

**Fig. 4 Total synthesis of (-)-sinoracutine (9).** Reagents and conditions: (a) PdCl$_2$ (0.1 equiv), O$_2$, NMP/H$_2$O = 8/1, 78%; (b) KOtBu (1.2 equiv), toluene/tBuOH = 20/1, 0 °C to rt, 91%; (c) DDQ (5.0 equiv), DCE, 50 °C, 81%; (d) LHMDS (1.5 equiv), NBS (1.2 equiv), −78 °C, THF, then TFA, DCM, rt, 72%. NMP N-methyl-2-pyrrolidinone, DDQ 2,3-dichloro-5,6-dicyano-1,4-benzoquinone, DCE 1,2-dichloroethane, NBS N-bromosuccinimide, TFA trifluoroacetic acid.

products. However, the same reaction carried out in a mixture of solvent (toluene/tBuOH = 20/1) in the presence of the same base furnished **26** in 91% isolated yield. Dehydrogenation of cyclopentenone **26** was carried out with an excess amount of 2,3-dichloro-5,6-dicyano-1,4-benzoquinone (DDQ) in 1,2-dichloroethane (DCE) to afford the highly π-conjugated tricycle **40** in 81% yield. At the outset of this work, we also synthesized diketones **41** and **42** by Wacker oxidation of **28** and **35**, respectively. However, neither of them underwent clean aldol condensation under either acidic or basic conditions due probably to the competitive Henry reaction among others.

Oxidation of cyclopentenone **40** to cyclopentenedione followed by intramolecular reductive amination was initially planned to forge the C5-N bond. However, the oxidation of **40** to α-dione failed under a variety of conditions. Therefore, regio- and diastereo-selective C5-bromination followed by intramolecular nucleophilic substitution reaction was envisaged to form the missing pyrrolidine ring[45,46]. Bromination of **40** with CuBr$_2$[47], phenyltrimethylammonium tribromide[48] and pyrrolidone hydrotribromide[49] afforded only the C8 brominated product without even a trace amount of the desired C5 isomer. The preferential C8 bromination is in fact understandable if one considers that C5, being a neopentyl carbon, is sterically less accessible. Assuming that this unfavorable steric bias could be overcome by rendering it more nucleophilic, we set out to perform the bromination on the preformed C5-enolate. Gratefully, deprotonation of **40** with LHMDS at −78 °C followed by addition of N-bromosuccinimide afforded α-bromoketone **43** in 78% yield as a single

diastereoisomer. Removal of the O-MOM and N-Boc groups and subsequent cyclization proceeded smoothly by simply stirring a dichloromethane solution of **43** in the presence of trifluoroacetic acid (TFA) to afford, after work-up with aqueous NaHCO$_3$ solution, the (-)-sinoracutine (**9**) in 90% yield. Finally, it was found that isolation of the brominated product **43** was unnecessary and compound **40** was transformed to (-)-sinoracutine (**9**) in a one-pot fashion in 72% overall yield. The spectroscopic data of the synthetic sample were identical with those of the natural product. The optical rotation was measured to be [α]$_D^{21}$ = −1035 (c 0.77, CHCl$_3$, 21 °C) and the ee of the synthetic compound was determined to be 94.4% by high-performance liquid chromatography (HPLC) analysis (see Supplementary Information). It is important to note that (-)-sinoracutine (**9**) is prone to racemization via either retro-Mannich/Mannich or retro Michael/Michael sequence as proposed by Trauner[34]. Since the last step leading to the natural product in our synthesis involved the removal of N-Boc function under acidic conditions, the retro-Mannich reaction might be effectively inhibited due to the protonation of the tertiary amine.

**Total synthesis of (-)-cepharatines A (5) and C (6).** To complete the total synthesis of (-)-cepharatines A (**5**) and C (**6**) (Fig. 5), one-carbon homologation of **38** to methyl ketone **44** was needed. After a number of unsuccessful trials, the desired transformation was realized in a one-pot fashion. Cross-metathesis of the terminal alkene **38** with commercially available isopropenylboronic acid pinacol ester (**45**) in the presence of Grubbs second-generation

**Fig. 5 Total synthesis of (-)-cepharatines A (5) and C (6).** Reagents and conditions: (a) isopropenylboronic acid pinacol ester (**45**, 10.0 equiv), Grubbs second-generation catalyst (0.1 equiv), DCM, 50 °C, then $NaBO_3.4H_2O$ (5.0 equiv), $THF/H_2O = 1/1$, rt, 62%; (b) 0.5 M NaOMe in MeOH, reflux, 70%; (c) LHMDS (2.0 equiv), MoOPH (3.0 equiv), THF, −78 °C to −20 °C, 74%; (d) DDQ (10.0 equiv), DCE, 50 °C, 68%; (e) DMP (1.3 equiv), DCM, 0 °C to rt, then TFA, 78%; (f) $H_2SO_4$ (1 M in MeOH)/$CH(OMe)_3$/MeOH = 0.1/1/1, 65 °C, 95%.

**Fig. 6 Total synthesis of (-)-cepharamine ( 4).** Reagents and conditions: (a) TMSOTf (3.0 equiv), 2,6-lutidine (5.0 equiv), DCM, 0 °C to rt, then TBAF (3.0 equiv), 0 °C, 77%; (b) KOtBu (2.0 equiv), $Et_2O/HCO_2Et = 10/1$, 0 °C to rt, then trimethylene dithiotosylate (1.0 equiv), AcOK (10.0 equiv), MeOH, reflux, 62%; (c) KOtBu (3.0 equiv), $Me_2SO_4$ (1.1 equiv), DMSO, rt, 85%; (d) PIFA (2.5 equiv), TFA (2.0 equiv), $MeCN/H_2O = 1/1$, then TFA, rt, 91%. TMSOTf trimethylsilyl trifluoromethanesulfonate, TBAF tetra-n-butylammonium fluoride, DMSO dimethyl sulfoxide, PIFA [bis(trifluoroacetoxy)iodo] benzene.

catalyst followed by oxidation of the resulting vinylboronate **46** with sodium perborate ($NaBO_3.4H_2O$) afforded diketone **44** in 62% yield[50,51]. Base-promoted intramolecular aldol condensation delivered the desired tricyclic enone **27** in 70% yield[52]. Since direct oxidation of **27** to dione under Riley conditions ($SeO_2$) led only to the decomposition of the starting material, an alternative two-step sequence via α-hydroxy ketone intermediate was pursued. Deprotonation of **27** with LHMDS followed by hydroxylation of the resulting lithium enolate with Vedejs' reagent [$MoO_5(Py)$ (HMPA)] generated the α-hydroxy ketone **47** as a single diastereoisomer in 74% yield[53]. We note that trapping of the enolate with Davis oxaziridine afforded the product **47** in much lower yield as did Rubottom oxidation. Dehydrogenation of **47** with DDQ in DCE at 50 °C afforded the highly conjugated intermediate **48** in 68% yield. Oxidation of α-hydroxy ketone to dione **49** followed by removal of the *N*-Boc and *O*-MOM groups and hemiaminal formation was expected to convert **49** to (-)-cepharatine A (**5**). Gratefully, this complex reaction sequence can be accomplished in an operationally simple one-pot fashion. Thus oxidation of α-hydroxy ketone **48** with DMP furnished dione **49** which, upon addition of TFA, was converted to the (-)-cepharatine A (**5**) in 78% overall yield. The formation of isomeric hemiaminal resulting from the formation of C7-N bond was not observed under these conditions. Following Reisman's procedure[32], (-)-cepharatine C (**6**) was obtained from (-)-cepharatine A (**5**) in excellent yield under mild acidic conditions. The physical and spectroscopic data of these two synthetic products were identical with those reported for the natural products.

**Total synthesis of (-)-cepharamine (4).** Total synthesis of (-)-cepharamine (**4**) is accomplished as shown in Fig. 6. Stirring a dichloromethane solution of **27** with TMSOTf and 2,6-lutidine followed by addition of TBAF afforded the aza-[4.4.3]-propellane structure **50** via a domino *N*-deprotection/intramolecular

aza-Michael addition sequence. To reach the targeted natural product, sequential selective functionalization of the C6 and the C8 carbons are needed and it was realized as follows. Regioselective C6-thioketalization of **50** was accomplished by reaction of its potassium enolate with trimethylene dithiotosylate[54]. Treatment of the resulting spirothioketal **51** with KOtBu followed by trapping of the resulting enolate with dimethyl sulfate provided methyl enol ether **52**. Finally, hydrolysis of dithioketal to ketone and removal of the *O*-MOM group was realized under acidic oxidative conditions (PIFA, TFA, rt)[55] to afford the (-)-cepharamine (**4**) in 91% yield.

## Discussion

We report in this paper a divergent synthetic strategy allowing the access to all three sub-classes of hasubanan family of alkaloids. Four representative members, namely, (-)-cepharamine (**4**), (-)-cepharatine A (**5**), (-)-cepharatine C (**6**), and (-)-sinoracutine (**9**) have been synthesized. The key feature of the present synthesis is the strategic generation of the quaternary C13 stereocenter with properly functionalized substituents allowing the subsequent construction of tricyclic enones. The C5, C6, and C14 of these intermediates are differentially activated rendering the selective C5-N, C6-N, and C14-N bond formation possible, hence three topologically different natural products. We believe that the present synthetic strategy is broadly applicable to members of this family of natural products and could be adopted to the synthesis of morphinan alkaloids by slightly modifying the reaction sequence.

## Methods

**General information.** Unless otherwise stated, starting materials were purchased from Aldrich and/or Fluka. Solvents were purchased in HPLC quality, degassed by purging thoroughly with nitrogen, and dried over activated molecular sieves of appropriate size. Alternatively, they were purged with argon and passed through alumina columns in a solvent purification system (Innovative Technology).

Conversion was monitored by thin layer chromatography (TLC) using Merck TLC silica gel 60 F254. Compounds were visualized by ultraviolet light at 254 nm and by dipping the plates in an ethanolic vanillin/sulfuric acid solution or an aqueous potassium permanganate solution followed by heating. Flash column chromatography was performed over silica gel (230–400 mesh). Nuclear magnetic resonance spectra were recorded on a Brüker AvanceIII-400, Brüker Avance-400 or Brüker DPX-400 spectrometer at room temperature, $^1$H frequency is at 400.13 MHz, and $^{13}$C frequency is at 100.62 MHz. Chemical shifts ($\delta$) were reported in parts per million (ppm) relative to residual solvent peaks rounded to the nearest 0.01 for proton and 0.1 for carbon (ref: CHCl$_3$ [$^1$H: 7.26, $^{13}$C: 77.16]. Coupling constants were reported in Hz to the nearest 0.1 Hz. Peak multiplicity was indicated as follows: s (singlet), d (doublet), t (triplet), q (quartet), m (multiplet), and br (broad). Attribution of peaks was done using the multiplicities and integrals of the peaks. Infrared spectra were recorded in a Jasco FT/IR-4100 spectrometer outfitted with a PIKE technology MIRacleTM ATR accessory as neat films compressed onto a Zinc Selenide window. The spectra were reported in cm$^{-1}$. The accurate masses were measured by the mass spectrometry service of the EPFL by electrospray ionization time of flight using a QTOF Ultima from Waters or APPI-FT-ICR using a linear ion trap Fourier transform ion cyclotron resonance mass spectrometer from Thermo Scientific. Melting points were measured using a Stuart SMP30.

## Data availability

The authors declare that the data supporting the findings of this study are available within the paper and the Supplementary Information, as well as from the authors upon request.

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

## Acknowledgements

We thank EPFL (Switzerland), the Swiss National Science Foundation (SNSF 20020_169077) for financial support.

## Author contributions

G.L., Q.W., and J.Z. conceived and designed the experiments. G.L. carried out the experiments. G.L., Q.W., and J.Z. interpreted the results and co-wrote the manuscript.

## Competing interests

The authors declare no competing interests.
