## [Peer Review File · Nature Communications]

REVIEWER COMMENTS

Reviewer #1 (Remarks to the Author):

In this manuscript, Zhu and co-workers describe an asymmetric synthesis of hasubanan alkaloids. In some senses, this synthesis is reminiscent of other morphinan and hasubanan alkaloids syntheses. Moreover, the key chiral intermediate 28 was prepared using You's method of "asymmetric dearomatization of beta-naphthols" (ref 38). In fact, the intermediate 28 is very similar to compound 3j of ref 38 and You demonstrated the transformation of this dearomatized product 38 to the propellane core structure of the hasubanan alkaloid. Nevertheless, I consider this work is interesting because it has an innovative feature and shows divergent total synthesis of all three sub-types of hasubanan alkaloids.

In general, the studies appear to have been conducted competently. The manuscript is well written and well organized. I consider the overall level of study is high enough for publication.

PS1: The structure of compound 19 is not correct.

PS2: In supporting material, please provide more detailed experimental information for chiral HPLC analysis of compound 28.

Reviewer #2 (Remarks to the Author):

The manuscript by Zhu and coworkers describes a unified strategy to hasubanan alkaloids featuring a catalytic enantioselective dearomatizative Michael addition for the construction of the key quaternary stereocenter in the common intermediate, which enabled the divergent synthesis of all three topologically different subclasses of this family of natural product through three different intramolecular C-N bond formations. Compared to the previous syntheses by other research groups, the current route does provide notable flexibility, which, this reviewer thinks, will draw attention from the chemical community. The results are described concisely, and additional observations as well as relevant comments were provided sufficiently. On the other hand, there is room for improvement in the introduction and the discussion sections. The part of the introduction on the precedents doesn't read well, and it will be desirable if the authors could provide a better context on the comparison between the tetra-substituted stereocenter formation strategies in the previous enantioselective total syntheses. The discussion section looks like a paraphrased abstract in the current form. The authors need to provide a bit more scientific discussion and/or perspectives. In addition, several errors are found in the figures and text. Overall, this reviewer recommends the acceptance after minor revision. The list of suggestions is given below.

Introduction

1. Although the authors tried to highlight the enantioselective constructions of tetra-substituted stereocenters in the precedents, the current form of this part looks like a plain listing. Moreover, the transitions between sentences are a bit abrupt. A more contextualized introduction needs to be given in order to guide the readers.
2. Reisman was able to access two sub-classes, which needs to be commented since this article emphasizes divergent synthesis.
3. Fig. 1a can be a bit confusing because it contains morphinan alkaloids while the caption is "representative structures of hasubanan alkaloids". It needs clarification that only B, C, and D are hasubanan.
4. The retrosynthetic scheme (Fig. 2) needs to be more informative. It will be easier to see the whole picture if the three key C-N bond-forming strategies are illustrated.
5. A reference is missing (line 36) for the synthesis of hasubanone by Ibuka (Tetrahedron Lett. 11, (1970) 4811).

Results

1. For the regio- and stereoselective C5-bromination of 40, initial attempts with various brominating reagents were unsuccessful. Eventually, the combined use of LHMDS/NBS afforded the desired bromide. What is the important factor? Is it the enhanced nucleophilicity resulting from deprotonation? Or, is NBS superior to other brominating reagents? It deserves more explanation.
2. The successful use of a mixed solvent system for the aldol condensation of 39 was described. Can the authors provide more details on the beneficial effect of the mixed solvent system?
3. The inhibition of the racemizing retro-Mannich process under acidic conditions is speculated at the end of the synthesis of 9. The same observation was made in the Trauner synthesis, and the same argument was also given. Thus, it needs to be either mentioned or cited.

Discussion

1. As mentioned above, it shouldn't be just a summary. Scientific discussion and/or perspectives need to be provided.

Errors and Typos

1. "quaternary stereocenter" in line 44 should be replaced with "tetrasubstituted stereocenter". Please note that a quaternary center means a carbon with four other carbon substituents.
2. "3,3" in line 48 needs square brackets (see the caption of Fig. 1f).
3. The dimethyl acetal in 18 should be a carbonyl group (Fig. 1d).
4. Two methoxy groups on the arene ring are missing in 19 (Fig. 1e).
5. The reference for You's procedure in line 84 is incorrect. It should be 38.
6. The prefixes "N-" and "O-" need to be italicized in several instances.
7. Several of the degree symbols in the figure captions are incorrectly formatted.
8. The sentence in line 134-136 seems grammatically incorrect.
9. The full name of DDQ was shown twice redundantly in Fig. 4 and 5.
10. The initials of the editor in ref 3 and the author in ref 9 need correction.
11. It seems that one resonance is missing in each of ¹³C NMR data of 47 and 48.

Ms Title: Unified Strategy to Hasubanan Alkaloids: Total Syntheses of (-)-Cepharamine, (-)-Sinoracutine, (-)-Cepharatines A and C” (MS N°: NCOOMS-20-37830)

Ms N°: NCOOMS-20-37830

Point-by-point responses to referees' comments

Reviewer #1:

Regarding “PS1: The structure of compound 19 is not correct”

It has been corrected in the revised version.

Regarding “PS2: In supporting material, please provide more detailed experimental information for chiral HPLC analysis of compound 28.

HPLC chromatogram of compound 28 together with the detailed analytic conditions were included in the revised SI (page 28). The HPLC analytic data were also included in the spectroscopic characterization of compound 28 (page 33).

Reviewer #2:

Introduction part

Regarding “1. Although the authors tried to highlight the enantioselective constructions of tetra-substituted stereocenters in the precedents, the current form of this part looks like a plain listing. Moreover, the transitions between sentences are a bit abrupt. A more contextualized introduction needs to be given in order to guide the readers.”

This part of the instruction has been modified. Some modifications are noted below: “A divergent synthesis of (-)-8-demethoxyrunanine and (-)-cepharatines A (), C (6) and D (8), belonging to the hasubanan subgroups B and C, respectively, were successfully realized from the common intermediate 18.” ...and ... “The common feature of these three approaches is the installation of the chiral -tertiary amine (C14) at the beginning of the synthesis. A completely different strategy implicating the generation of C13 quaternary carbon stereocenter was designed by Trauner and co-workers for the total synthesis of (-)-sinoracutine (9).³⁴ Thus, oxa-Michael addition of 22 to phenyl vinyl sulfoxide (23) afforded 24 (NaH, THF, then KH, 0 °C to rt) which, upon heating in 1,2-dichlorobenzene (NaHCO₃, 176 °C), underwent *syn*-elimination of sulfenic acid to afford the chiral allylic enol ether. The Claisen rearrangement of the latter furnished then the aldehyde 25 with perfect chirality transfer from C6 to C13 (Fig. 1f).”

Regarding “2. Reisman was able to access two sub-classes, which needs to be commented since this article emphasizes divergent synthesis.

This point has been stressed in the revised version. It reads: “Concurrently, Reisman and co-workers designed a general strategy involving a diastereoselective addition of Grignard reagent 16 to chiral -tert-butanesulfinimine 17 for the generation of C14 tetrasubstituted stereocenter in 18 (Fig. 1d).³² A divergent synthesis of (-)-8-demethoxyrunanine and (-)-cepharatines A (), C (6) and D (8), belonging to the hasubanan subgroups B and C, respectively, were successfully realized from the common intermediate 18.”

Regarding “3. Fig. 1a can be a bit confusing because it contains morphinan alkaloids while the caption is “representative structures of hasubanan alkaloids”. It needs clarification that only B, C, and D are hasubanan.”

The caption **a** has been changed to “**a Representative skeletons of morphinan (A) and hasubanan alkaloids (B, C and D);**”

Regarding “4. The retrosynthetic scheme (Fig. 2) needs to be more informative. It will be easier to see the whole picture if the three key C-N bond-forming strategies are illustrated.”

The Figure 2 has been re-drawn.

Regarding “5. A reference is missing (line 36) for the synthesis of hasubanone by Ibuka (Tetrahedron Lett. 11, (1970) 4811).”

It has been added as reference 12. It reads: 12. Ibuka, T., Tanaka, K. & Inubushi, Y. The total synthesis of dl-hasubanone. *Tetrahedron Lett.* **11, 4811 (1970).**

Results part

Regarding “1. For the regio- and stereoselective C5-bromination of 40, initial attempts with various brominating reagents were unsuccessful. Eventually, the combined use of LHMDs/NBS afforded the desired bromide. What is the important factor? Is it the enhanced nucleophilicity resulting from deprotonation? Or, is NBS superior to other brominating reagents? It deserves more explanation.”

Following sentence was added. It reads: “**The preferential C8 bromination is in fact understandable if one considers that C5, being a neopentyl carbon, is sterically less accessible. Assuming that this unfavorable steric bias could be overcome by rendering it more nucleophilic, we set out to perform the bromination on the preformed C5-enolate.**”

Regarding “2. The successful use of a mixed solvent system for the aldol condensation of 39 was described. Can the authors provide more details on the beneficial effect of the mixed solvent system?”

Following sentence was added and it reads: “**Performing the intramolecular aldol condensation of 39 in *t*BuOH in the presence of potassium *tert*-butoxide (*t*BuOK) proceeded slowly affording tricyclic enone 26 in low yield together with multiple unidentified products. However, the same reaction carried out in a mixture of solvent (toluene/*t*BuOH = 20/1) in the presence of the same base furnished 26 in 91% isolated yield.**”

Regarding “The inhibition of the racemizing retro-Mannich process under acidic conditions is speculated at the end of the synthesis of 9. The same observation was made in the Trauner synthesis, and the same argument was also given. Thus, it needs to be either mentioned or cited.”

This reference has been re-cited.

Discussion part

Regarding “1. As mentioned above, it shouldn’t be just a summary. Scientific discussion and/or perspectives need to be provided.

It has been re-written and it reads: “We report in this paper a divergent synthetic strategy allowing for the first time the access to all three sub-classes of hasubanan family of alkaloids. Four representative members, namely (-)-cepharamine (4), (-)-cepharatine A (5), (-)-cepharatine C (6) and (-)-sinoracutine (9) have been synthesized. The key feature of the present synthesis is the strategic generation of the quaternary C13 stereocenter with properly functionalized substituents allowing the subsequent construction of tricyclic enones. The C5, C6 and C14 of these intermediates are differentially activated rendering the selective C5-N, C6-N and C14-N bond formation possible, hence three topologically different natural products. We believe that the present synthetic strategy is broadly applicable to members of this family of natural products and could be adopted to the synthesis of morphinan alkaloids by slightly modifying the reaction sequence.”

Regarding “Errors and Typos”

All the typos and grammatical errors have been corrected as suggested.

We thank again the referees for their thoughtful comments. We believe that the quality of the manuscript has been significantly improved after having addressed all these key questions raised by the reviewers.

Best regards
Yours sincerely,

Prof. Jieping Zhu, PhD

REVIEWERS' COMMENTS

Reviewer #2 (Remarks to the Author):

All the suggestions have been well addressed in the revised manuscript by Zhu and coworkers. The introduction is easy to read, and the authors' divergent strategy is clearly illustrated. A few informative details have been added, and the conclusion provides some perspectives. This reviewer believes that this manuscript is ready for publication.